ꝺ | **Open Peer Review** | Clinical Microbiology | Research Article

# Identification of neutrophil-related genes associated with the severity of RSV infection

**Daiyang Zhang,**[1,2] **Hewei Zhang,**[1] **Xin Zhang,**[1] **Lei Yin,**[1] **Xuejun Shao,**[1] **Shenghao Hua**[1]

**ABSTRACT** Respiratory syncytial virus (RSV) infection is a major cause of acute respiratory illness in infants and young children, leading to significant health and economic challenges. This study aimed to identify key neutrophil-related genes associated with the severity of RSV infection and to explore their potential as biomarkers and therapeutic targets. The GSE188427 and GSE246622 data sets in the Gene Expression Omnibus (GEO) were selected to identify key neutrophil-related genes. We conducted Gene Ontology (GO) and Kyoto Encyclopedia of Genes and Genomes (KEGG) analyses to identify enriched pathways. The ROC curve was applied to evaluate the diagnostic effectiveness of key genes. Clinical samples were used to validate these genes. Finally, we constructed networks of miRNA–gene and drug–gene interactions. Integrated bioinformatics analysis revealed 10 hub genes. Further validation revealed that most of them exhibited increased expression as symptoms worsened. In addition, the expression of these 10 hub genes decreased as the symptoms improved. ROC curve analyses indicated that BPI and ELANE effectively predict hospitalization needs for outpatients, whereas ARG1 and PADI4 are associated with severe disease progression. Validation with clinical samples confirmed that the expression levels of BPI, ELANE, ARG1, and PADI4 were positively associated with RSV severity. This study identified potential therapeutic targets, paving the way for future drug development and improved clinical management of RSV-related illnesses. An enhanced understanding of these biomarkers may facilitate earlier intervention and more tailored treatment strategies for affected patients.

**IMPORTANCE** Given the significant burden of respiratory syncytial virus (RSV) in infants and young children, our work addresses an important aspect of respiratory disease by identifying molecular markers that correlate with disease progression and recovery. This study investigates the relationship between specific neutrophil-related genes and the severity of RSV infection, highlighting their promise as biomarkers to aid clinical assessment and management.

**KEYWORDS** neutrophil-related genes, respiratory syncytial virus, biomarkers

Respiratory syncytial virus (RSV) is the primary cause of acute respiratory infections in children under 5 years of age, leading to numerous hospitalizations and substantial healthcare costs annually (1). Research shows that RSV can cause various clinical symptoms, ranging from mild upper respiratory infections to severe pneumonia requiring hospitalization (2). Current RSV treatment strategies mainly emphasize supportive care and a few antiviral agents; however, these methods often fail to address early detection and risk assessment for severe cases (3). This situation highlights the urgent need for improved diagnostic and treatment methods to reduce the impact of severe RSV infection.

Recent studies have increasingly demonstrated that neutrophils are associated with severe RSV infection (4–6). Furthermore, compared with those in rhinovirus and influenza

**Peer Reviewer** Yoshitaka Kimura, Teikyo Daigaku, Tokyo, Japan

Address correspondence to Shenghao Hua, huashenghao@suda.edu.cn.

The authors declare no conflict of interest.

infection, the expression levels of neutrophil-associated genes are significantly increased in severe RSV infection (7). However, the specific genes related to neutrophil responses during severe RSV infection are not well understood. This gap in knowledge indicates significant potential for progress in identifying biomarkers that could predict disease severity and guide treatment strategies. Therefore, the primary objective of this research was to identify key neutrophil-related genes associated with RSV infection and to validate their roles in determining disease severity and prognosis. We aimed to create interaction networks that include these genes, potential drug candidates, and micro-RNAs, establishing a foundation for future treatment development. This comprehensive approach aims to both enhance our understanding of RSV pathogenesis and improve the clinical management of RSV infections through the identification of predictive biomarkers and therapeutic targets.

## MATERIALS AND METHODS

### Data sources and processing

The GSE188427 and GSE246622 data sets were downloaded from the Gene Expression Integrated Database (GEO) (http://www.ncbi.nlm.nih.gov/geo/). The identification of differentially expressed genes (DEGs) was conducted via data from GSE188427, which included 51 healthy children, 37 RSV-infected outpatients on day 1, and 110 RSV-infected inpatients on day 1. For the validation of hub genes, data from GSE246622 included 36 RSV-infected outpatients with mild disease, 40 RSV-infected inpatients with moderate disease, and 18 RSV-infected inpatients with severe disease. Additionally, to assess the role of hub genes in predicting the prognosis of RSV infection, data from GSE188427 included 110 RSV-infected inpatients on days 1, 69 on day 30, and 55 on day 180. All the samples were derived from *Homo sapiens* via two platforms: the GPL25336 [Clariom_S_Human_HT] Affymetrix Clariom S Pico Assay HT and the GPL31262 [GO_Screen_hu] Affymetrix human GO_Screen_hu Assay. Neutrophil-related genes were downloaded from Genecards (https://www.genecards.org/).

### Identification of DEGs

GEO2R was used to identify DEGs. The "ggplot2" and "pheatmap" packages in R.4.2.1 were used to construct volcano plots and heatmaps of the DEGs. We defined significant DEGs as those with an adjusted *P*-value < 0.05 and |log2FC| > 0.5.

### Functional and pathway enrichment analyses

Gene Ontology (GO) and Kyoto Encyclopedia of Genes and Genomes (KEGG) enrichment analyses were conducted for gene intersections via the R package "clusterProfiler" (8). The visualization was implemented via the R packages "ggplot2," "igraph," and "ggraph." Significant results were determined with an adjusted *P*-value of <0.05. If >10 terms were identified, then the top 10 terms were visualized via the methods described.

### Gene set enrichment analysis

Gene Set Enrichment Analysis (GSEA) was performed using GSEA 4.1.0 software (Broad Institute, Cambridge, MA, USA) to analyze functional enrichment patterns in genome-wide gene expression. The reference gene set database was h.all.v2026.1.Hs.symbols.gmt (Hallmark gene sets, downloaded from the Broad Institute's Molecular Signatures Database, MSigDB).

## Protein–protein interaction (PPI) analysis and identification of hub genes

PPI networks were extracted via the STRING database (https://cn.string-db.org/) (9) and imported into Cytoscape 3.9.1 to visualize the PPI network of DEGs (10). Hub genes were identified via the "Cytohubba" plug-in (11).

## Validation with clinical samples

To clarify the severity of RSV disease, we implemented a clinical scoring system comprising five evaluative criteria: respiratory rate, auscultation results, transcutaneous O2 saturation, retractions, and activity level, which were primarily assessed through feeding challenges (12). Each criterion is allocated a score ranging from 0 (denoting normalcy) to 3 (signifying marked abnormality). Using this scoring framework, patients were classified as having mild (0–5), moderate (6–10), or severe (11–15) RSV lower respiratory tract infections (LRTIs). Peripheral blood samples were collected from 31 outpatients with mild RSV infections, 27 hospitalized children with moderate RSV infections, and 15 hospitalized children with severe RSV infections on the day they visited the Children's Hospital of Soochow University. These children exhibited typical respiratory symptoms and had positive nucleic acid tests for respiratory syncytial virus in their sputum or throat swabs. They were not diagnosed with hematological diseases, malignant tumors, or congenital heart disease.

Total RNA was extracted from leucocytes by using TRIzol reagent. cDNA was prepared by reverse transcription with oligo(dT) from total RNA extraction. qRT–PCR for ARG1, ELANE, BPI, PADI4, and a reference gene (HPRT) was performed on an ABI 7500 instrument with a SYBR Green Mastermix kit. Each target gene expression result was then normalized relative to that of HPRT. The sequences of primers used were as follows (Table 1).

## ROC curve

The R package "pROC" in R.4.2.1 was used for plotting the ROC curves. The visualization was implemented via the R package "ggplot2."

## Construction of the drug–gene interaction network

The Drug–Gene Interaction Database (DGIdb) version 5.0.8 (https://www.dgidb.org) serves as an online repository that amalgamates data from various drug–gene interaction databases (13). An analysis was conducted on potential pharmaceuticals or compounds associated with the hub genes through DGIdb. The resulting drug–gene interaction network was depicted via Cytoscape.

## Construction of the miRNA–gene interaction network

The miRNA–gene network was constructed via miRWalk (http://mirwalk.umm.uni-heidelberg.de/), a platform that houses predictive data generated by machine learning algorithms alongside experimentally validated miRNA–gene interactions. miRWalk facilitated the identification of miRNAs that might modulate hub genes, with selections made for miRNAs linked to four or more hub genes boasting a score exceeding 0.9. The final interaction network was illustrated via Cytoscape.

**TABLE 1** Primer sequences of genes

| Gene | Forward primer (5′–3′) | Reverse primer (5′–3′) |
| --- | --- | --- |
| ARG1 | ACTTAAAGAACAAGAGTGTGATGTG | CATGGCCAGAGATGCTTCCA |
| ELANE | CGTGGCGAATGTAAACGTCC | TTTTCGAAGATGCGCTGCAC |
| BPI | GCTTCAGCCTCACCAGAACT | CGAAGAAGTTTGCAAGGGGC |
| PADI4 | TTCTGCCATGGACTGCGAG | GGGTATTCCTTGCCCCTGAC |
| HPRT | AGGACTGAACGTCTTGCTCG | ATCCAACACTTCGTGGGGTC |

## Statistical analysis

All the statistical analyses were conducted via R statistical software (version 4.2.1, http://r-project.org/). One-way ANOVA was applied to assess normally distributed continuous data. Every statistical examination was two-sided, with a significance threshold set at $P \leq 0.05$ deemed statistically relevant.

## RESULTS

### DEGs and functional/pathway enrichment analysis

To identify neutrophil-related genes linked to RSV severity, we first analyzed the GSE188427 data set encompassing healthy controls, RSV-infected outpatients, and RSV-infected inpatients. Using GEO2R with thresholds of adjusted $P < 0.05$ and |log2FC| > 0.5, we detected 392 upregulated and 98 downregulated genes in RSV-infected inpatients versus controls, and 459 upregulated and 134 downregulated genes in RSV-infected outpatients versus controls (Fig. 1a and b). Venn diagram analysis intersecting these DEGs with neutrophil-related genes retrieved from Genecards identified 117 neutrophil-associated DEGs exclusive to hospitalized pediatric RSV patients and not detected in outpatients (Fig. 1c). Heatmaps then visualized their distinct expression patterns across study groups (Fig. 1d), suggesting that these genes may be uniquely implicated in severe disease phenotypes. Subsequently, we performed GO and KEGG enrichment analyses on these 117 neutrophil-associated DEGs to delineate their biological functions and associated pathways in RSV pathogenesis. Gene Ontology (GO) Biological Process (BP) enrichment revealed significant associations, including the negative regulation of cytokine production, response to lipopolysaccharide, negative regulation of interleukin-6 production, and defense response to fungus (adjusted $P < 0.05$; Fig. 2a). These findings align with the well-established role of neutrophils in modulating inflammatory responses during viral respiratory infections and indicate that these neutrophil-associated DEGs may contribute to excessive or dysregulated inflammation in severe RSV. Cellular Component (CC) analysis showed enrichment in the cytoplasmic vesicle lumen, secretory granule lumen, and specific granules (Fig. 2a), which are subcellular compartments critical for neutrophil degranulation. This process is a key mechanism by which neutrophils release antimicrobial factors and proinflammatory mediators during infection. Molecular Function (MF) analysis indicated associations with immune receptor activity, MHC protein complex binding, and lipopolysaccharide binding (Fig. 2a), reflecting the involvement of these neutrophil-associated DEGs in neutrophil-mediated pathogen recognition and antigen presentation, processes central to antiviral immunity. Kyoto Encyclopedia of Genes and Genomes (KEGG) pathway enrichment analysis further revealed that the 117 neutrophil-associated DEGs were significantly enriched in pathways, including hematopoietic cell lineage, antigen processing and presentation, graft-versus-host disease, asthma, and drug metabolism-other enzymes (Fig. 2b). Enrichment in the hematopoietic cell lineage pathway supports the role of these genes in regulating neutrophil development and activation, processes known to be dysregulated in severe RSV infection. Meanwhile, enrichment in antigen processing and presentation underscores their potential involvement in modulating adaptive immune responses to RSV. Collectively, these functional and pathway analyses demonstrate that the 117 neutrophil-associated DEGs are biologically relevant to neutrophil-mediated inflammation and antiviral responses, highlighting their potential role as key mediators of RSV disease severity.

### Gene set enrichment analysis

To further elucidate the specific molecular pathways uniquely present in hospitalized pediatric patients, we performed GSEA on peripheral blood transcriptome data comparing RSV outpatients and healthy children, as well as RSV inpatients and healthy children. We found that four pathways—including coagulation, adipogenesis, reactive oxygen species pathway, and estrogen response pathway—exhibited significant

transcriptomic differences only in RSV hospitalized children (Fig. 2c). The reactive oxygen species pathway exhibited perturbations indicative of neutrophil-induced oxidative stress, while the coagulation pathway was affected by inflammation-coagulation interactions due to neutrophil granule protein release. Additionally, the adipogenesis pathway was reprogrammed by proinflammatory cytokines from activated neutrophils, thereby influencing neutrophil energy metabolism and function. The estrogen late response pathway showed transcriptional changes due to neutrophil-mediated

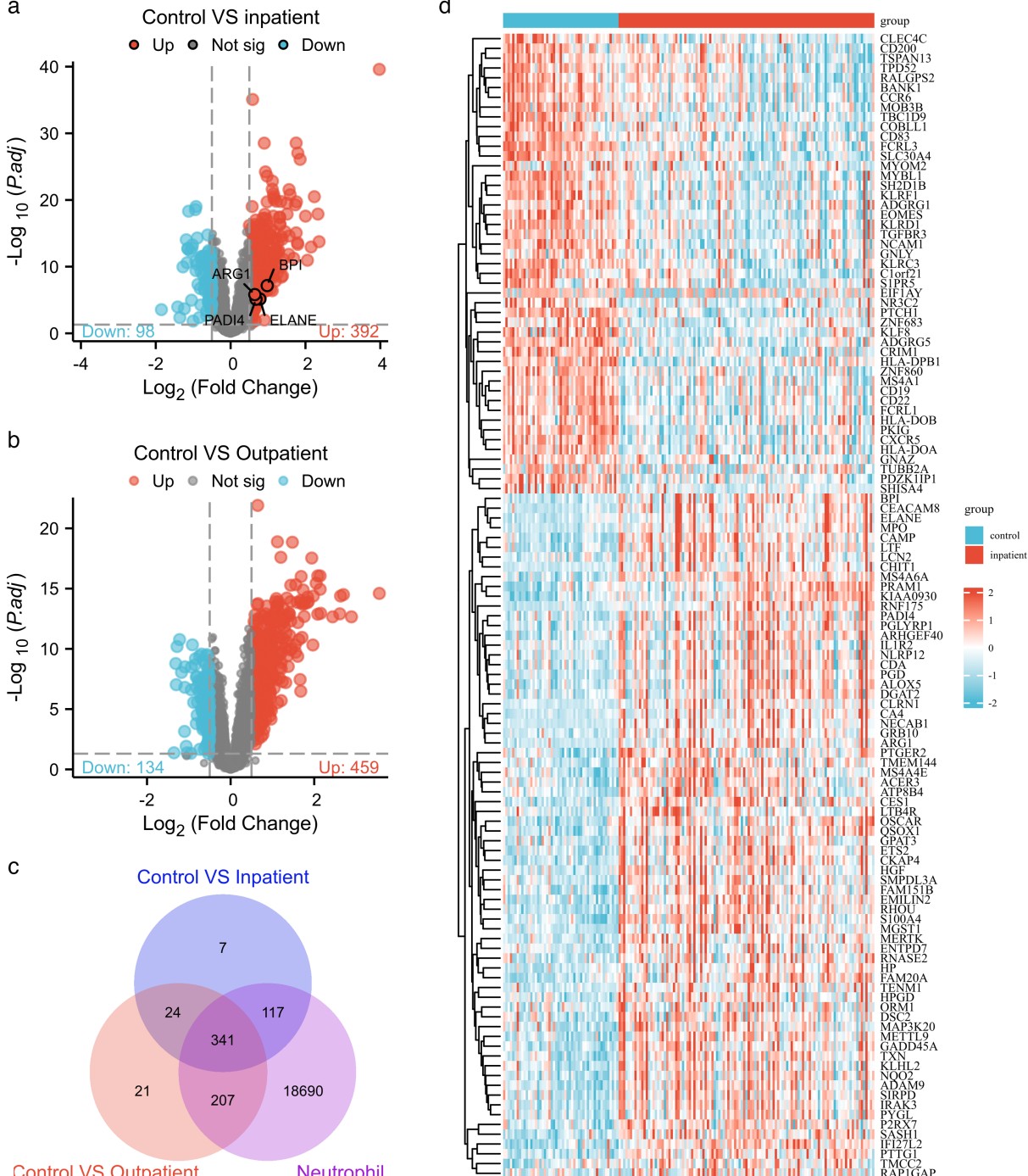

**FIG 1** Identification of DEGs. (a and b) Volcano plots showing the DEGs. Red and blue dots represent upregulated and downregulated genes, respectively, and the parameters were set as |log2 fold change (FC)| > 0.5 and adjusted *P*-value < 0.05. (c) Venn diagram presenting a combination of up-/downregulated genes and neutrophil-related genes. (d) Heatmaps showing the 117 DEGs in GSE188427.

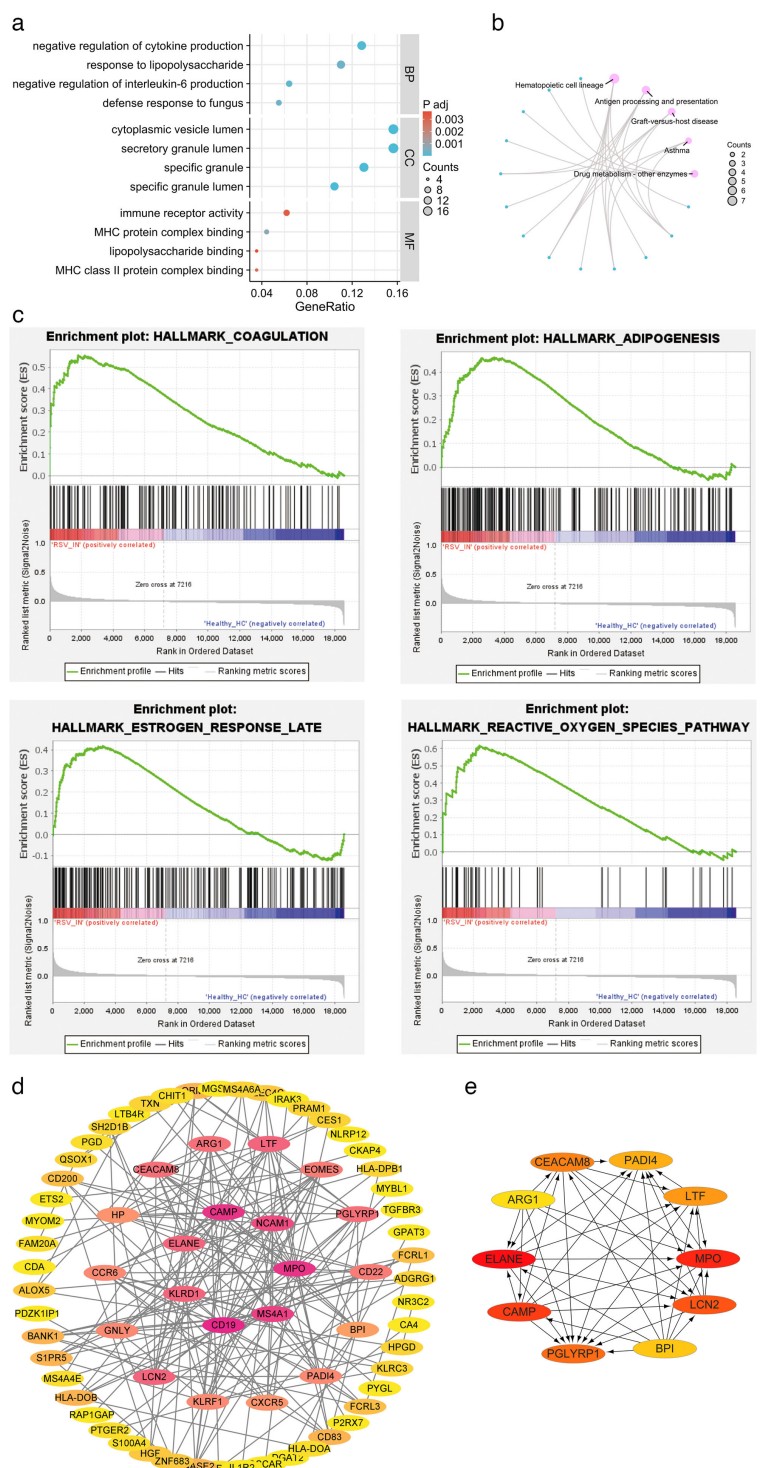

**FIG 2** Functional/pathway enrichment analyses and PPI network of DEGs. (a) Bar chart showing GO enrichment analysis of DEGs. (b) Cnetplot visualizing the conjunction between genes and the KEGG pathway. (c) GSEA using reference gene sets from HALLMARK database. (d) STRING database analysis of DEGs. Cytoscape was used to plot the PPI network, with colors reflecting the score rank calculated by CytoNCA (red for a higher degree, yellow for a lower degree). (e) Hub genes (10 in total) identified from Cytohubba.

inflammation. Meanwhile, the dysregulation of these four pathways enhanced neutrophil activation, creating a feedback loop. This indicates that excessive neutrophil activation drives RSV infection progression to severe disease.

## PPI network and hub genes

These 117 neutrophil-associated DEGs were scrutinized via the STRING database, resulting in 117 nodes and 196 edges. The PPI network was illustrated via Cytoscape (Fig. 2d). From this analysis, we identified 10 pivotal hub genes: CEACAM8, PADI4, LTF, MPO, LCN2, BPI, PGLYRP1, CAMP, ELANE, and ARG1 (Fig. 2e).

## Validation of hub gene expression in GSE246622

We analyzed the expression of these 10 hub genes in the GSE246622 data set, which included an outpatient group of RSV-infected children with mild symptoms (*n* = 36), an inpatient group of RSV-infected children with moderate symptoms (*n* = 40), and an inpatient group of RSV-infected children with severe symptoms (*n* = 18). The results indicated that, with the exception of MPO and CAMP, the expression of the remaining eight genes increased with symptom severity (Fig. 3a). Additionally, we analyzed the expression of these 10 hub genes from the GSE188427 data set in the Day 1, Day 30, and Day 180 groups of hospitalized RSV children. The results revealed that on Day 30, the

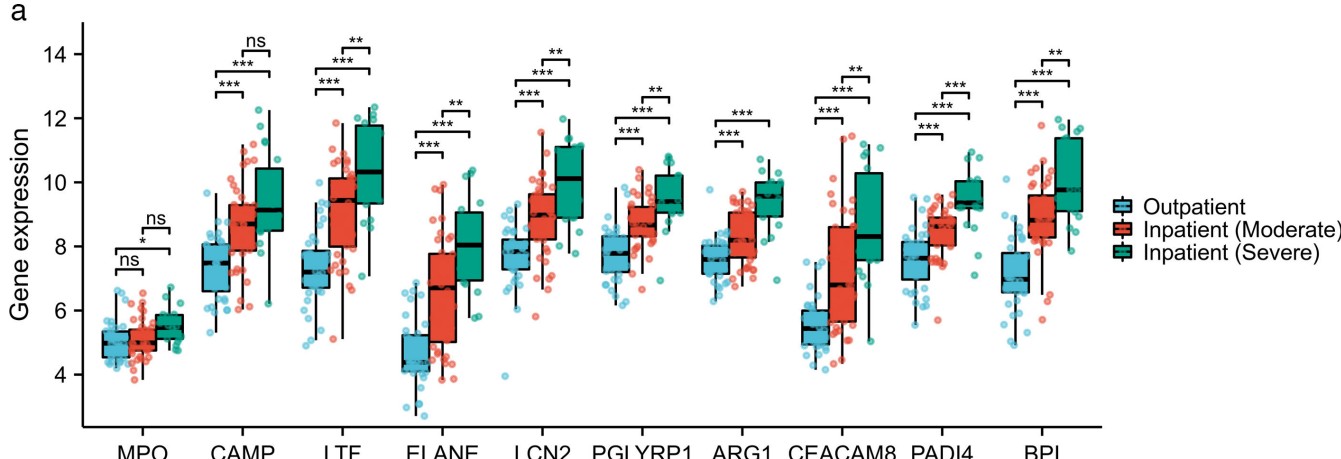

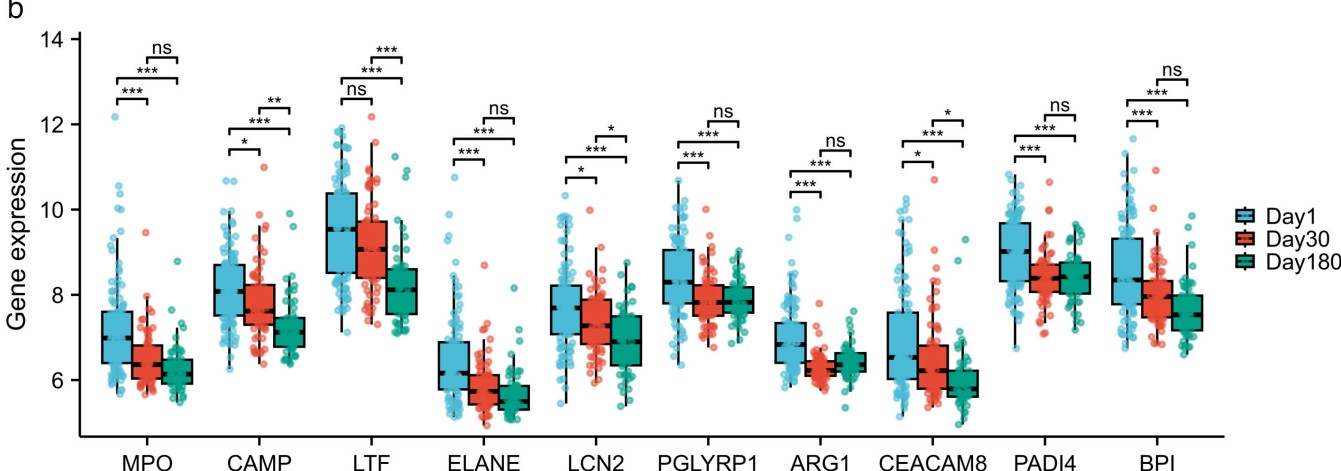

**FIG 3** Validation of hub gene expression in GSE246622 and the predictive prognostic role of the hub genes. (a) Expression of the hub genes in the GSE246622 data set. (b) Expression of hub genes on days 1, 30, and 180 after RSV infection. The upper line, middle line, and lower line in the box-and-whisker plots represent the upper quartile, mean, and lower quartile, respectively. *P < 0.05; **P < 0.01; ***P < 0.001; ****P < 0.0001; ns, not significant.

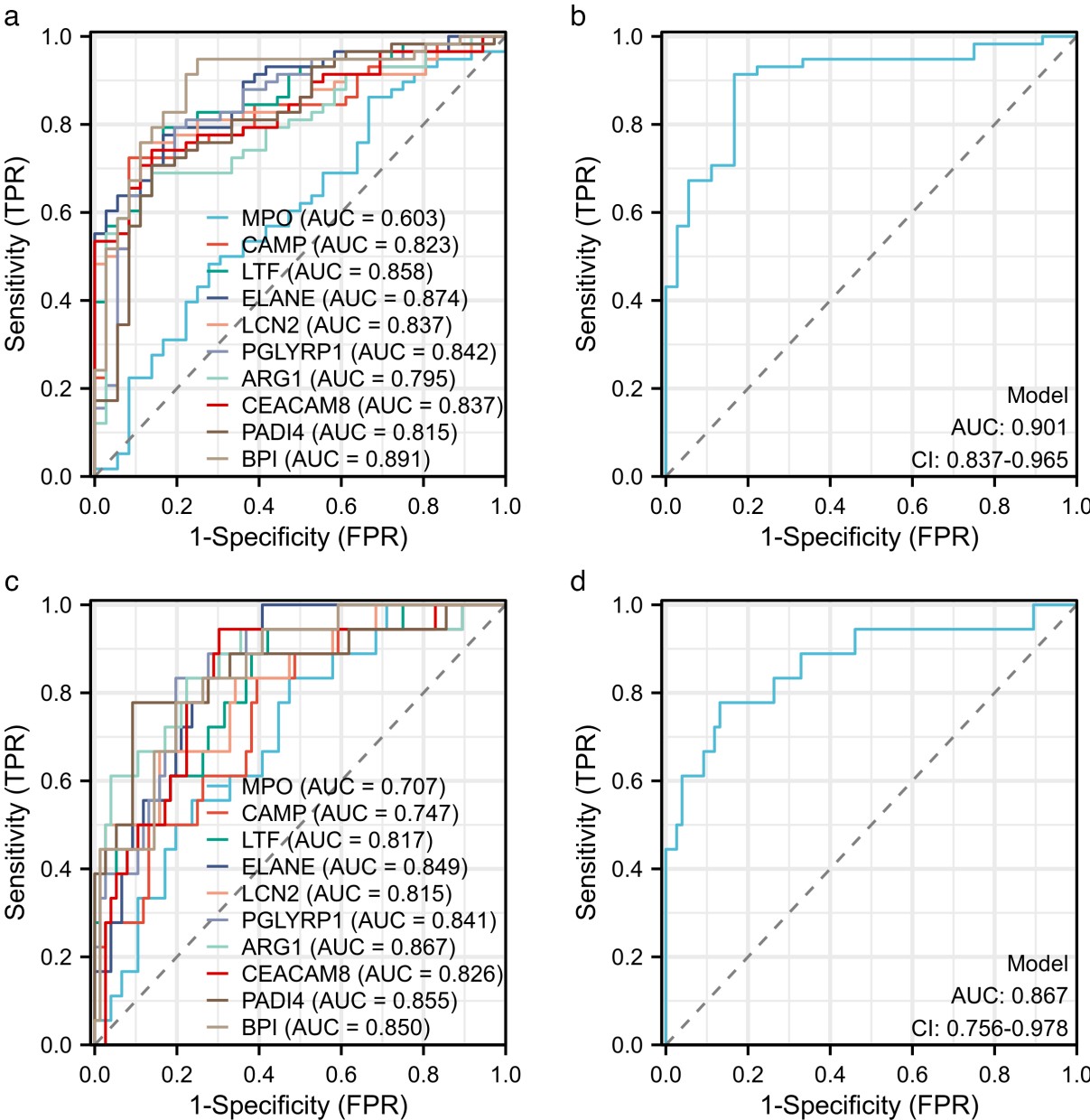

**FIG 4** Diagnostic efficacy of the hub genes according to the ROC curve analysis. (a) ROC curve of the hub genes for predicting outpatient RSV-positive children who need hospitalization. (b) ROC curve of the combination of the BPI and ELANE for the prediction of outpatient RSV-positive children who need hospitalization. (c) ROC curve of the hub genes for predicting that hospitalized children with RSV will progress to severe cases. (d) ROC curve of combining ARG1 and PADI4 for predicting that hospitalized children with RSV will progress to severe cases.

expression of all genes except LFT decreased as symptoms improved, and by Day 180, the expression of all genes significantly decreased (Fig. 3b).

## Diagnostic efficacy of the hub genes

We conducted ROC curve analysis using the data set GSE24662 to evaluate how effectively hub genes can be used to determine which outpatient RSV-positive children require hospitalization and which hospitalized RSV-positive children are at risk of developing severe cases. We plotted ROC curves using hub genes from 36 outpatient children and 58 hospitalized children, as well as from 18 critically ill children and 76 non-critically ill children. The results revealed that the genes BPI and ELANE were the

most effective at identifying outpatient RSV-positive children who needed hospitalization (Fig. 4a). Consequently, we combined these two genes in a further ROC curve analysis, which revealed an AUC of 0.901 (Fig. 4b). The genes ARG1 and PADI4 were the most highly predictive genes for severe RSV, so we combined them in another ROC curve analysis, which revealed an AUC of 0.867 (Fig. 4c and d).

## Clinical validation

We collected peripheral blood samples from 31 outpatients with mild RSV, 27 hospitalized patients with moderate RSV, and 15 hospitalized patients with severe RSV. qRT−PCR was used to detect the relative expression of BPI, ELANE, ARG1, and PADI4. The time point for blood sampling is during the early stage of the disease, that is, 2–3 days after the child has symptoms of respiratory tract infection. The baseline characteristics among the groups are shown in Table 2. The results indicated that only ELANE did not significantly differ between the mild and moderate case groups (Fig. 5). This finding suggests that as the symptoms of RSV infection worsen, the expression levels of BPI, ARG1, and PADI4 increase.

## Networks of drug−gene and miRNA−gene interactions

ARG1, ELANE, BPI, and PADI4 were identified as potential drug targets in the interaction network (Fig. 6a), whereas nine miRNAs that may regulate three of these hub genes were selected (Fig. 6b).

## DISCUSSION

RSV is the leading cause of lower respiratory tract infections in children. Some affected children might only show mild symptoms, whereas others may progress to severe cases that require hospitalization or even admission to the intensive care unit (ICU) (2). Consequently, accurately identifying severe RSV patients is key to improving treatment outcomes and patient prognosis. Currently, several biomarkers, such as antibody levels, immune cell changes, microbial exposures, the host immune response, and HLA-DR$^{Low}$ monocyte subsets, are used to distinguish infants with severe RSV disease (5, 7, 14). However, no studies have investigated specific neutrophil-related genes associated with the severity of RSV infection.

This study, on the basis of the GSE188427 data set, explored biomarkers for severe RSV infection in children. We identified 10 important hub genes that may serve as potential targets for severe RSV treatment in the future. In the validation study of hub gene expression, we found that the expression of most hub genes was positively correlated with the severity of symptoms. ROC curve analysis revealed that the BPI and ELANE genes performed best in predicting hospitalization needs in outpatient RSV patients, whereas the ARG1 and PADI4 genes had significant value in predicting severe patients. Previous studies have shown that BPI and ARG-1 are associated with severe

**TABLE 2** The baseline characteristics among the groups

| Characteristic | Disease severity | | | P |
| --- | --- | --- | --- | --- |
| | Mild (*n* = 31) | Moderate (*n* = 27) | Severe (*n* = 15) | |
| Sex, no. (%) | | | | |
| Female | 9 (29.0) | 10 (37.0) | 7 (46.7) | 0.49 |
| Male | 22 (71.0) | 17 (63.0) | 8 (53.3) | |
| Age, median (IQR), mo | 39.1 (24.0–48.0) | 19.7 (6.5–36.0) | 10.8 (2.5–11.0) | <0.05 |
| Age, no. (%) | | | | |
| <3 mo | 0 | 1 (3.7) | 4 (26.7) | |
| 3 to <6 mo | 1 (3.2) | 4 (14.8) | 2 (13.3) | |
| 6 to <12 mo | 1 (3.2) | 6 (22.2) | 5 (33.3) | |
| 12 to <24 mo | 3 (9.7) | 3 (11.1) | 1 (6.7) | |
| 2 to <5 y | 19 (61.3) | 13 (48.1) | 3 (20.0) | |
| 5 to <18 y | 7 (22.6) | 0 | 0 | |

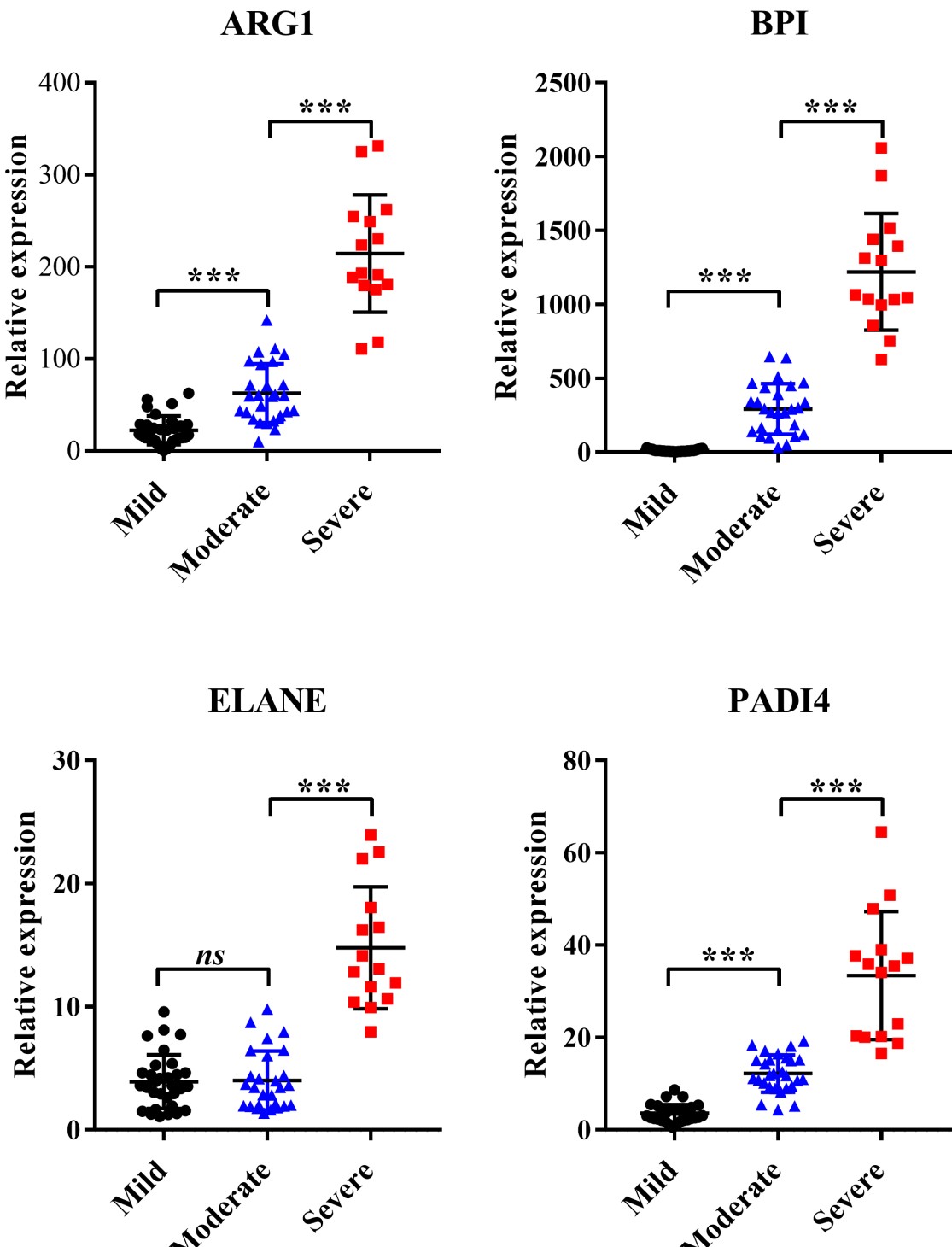

**FIG 5** Validation of the hub genes in clinical samples. Detection of the expression of ARG1, ELANE, BPI, and PADI4 in the peripheral blood of RSV-infected children with varying degrees of symptoms via qRT−PCR. The bars represent mean ± SD. ****$P < 0.001$; ns, not significant.

COVID-19, whereas BPI and ELANE are key candidate biomarkers for severe influenza infection (15–19). This suggests that severe respiratory infections may share certain commonalities in some respects. Moreover, our study explored the interactions of genes with drugs and miRNAs and revealed that these genes may provide clues for new drug development and regulatory mechanisms. In summary, this study not only provides

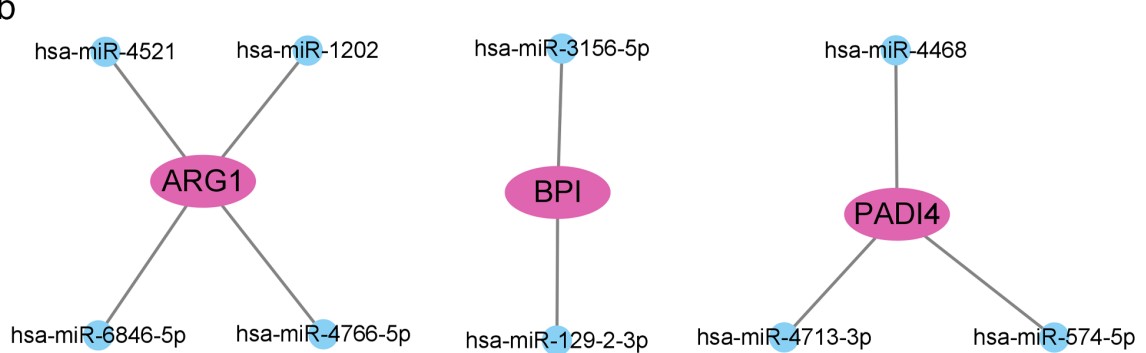

**FIG 6** Networks of drug–gene and miRNA–gene interactions. (a) Network of drug–gene interactions. The blue nodes represent drugs or molecular compounds, and the orange nodes represent hub genes related to drugs. (b) Network of miRNA–gene interactions. The purple nodes represent hub genes, and the blue nodes represent miRNAs.

new insights into biomarkers for severe RSV infection but also offers potential biological targets for early intervention in clinical practice.

The limitations of this study include the small sample size, particularly in the severe RSV patient group, which included only 15 samples, potentially resulting in less robust statistical analysis results. Additionally, although we validated the expression of the hub genes through various methods, further validation in larger clinical cohorts is needed to increase the external validity of the results. Furthermore, while we explored the interactions of genes with drugs and miRNAs, our understanding of their biological

mechanisms remains insufficient. Moreover, we did not verify the concentrations of BPI, ARG1, PADI4, or ELANE at the protein level in children with varying degrees of RSV infection severity. These limitations suggest that future research should involve broader sample sources and more comprehensive biomarker exploration to improve prognostic assessment capabilities for severe RSV infections.

In conclusion, this study identified key biomarkers associated with severe RSV infection through DEG analysis and clinical validation. These biomarkers have the potential to distinguish patients with mild RSV infection from those with severe RSV infection. In particular, genes, such as BPI, ELANE, ARG1, and PADI4, can serve as both predictors of hospital admission necessity and early warning indicators for severe disease progression. These findings provide new insights for the early identification and intervention of severe RSV infections, laying the groundwork for future treatment strategies.

## Article summary

This study identified 10 neutrophil-related genes linked to RSV severity, with BPI and ELANE predicting hospitalization needs and ARG1 and PADI4 indicating severe progression.

## What's known on this subject

Neutrophils are closely related to severe RSV infection.

## What this study adds

We have identified specific neutrophil genes that are closely linked to severe RSV infection.

## ACKNOWLEDGMENTS

We thank all the participants in the study.

This work was supported by the Youth Science and Technology Program of Suzhou Municipal Health Commission (KJXW2023025) and the Suzhou Science and Technology Development Plan (SYW2024107).

The Suzhou Municipal Health Commission had no role in the design and conduct of the study.

Z.D. and H.S. conceptualized and designed the study, drafted the initial manuscript, and reviewed and revised the manuscript. S.X. and Z.H. carried out the initial analyses, and reviewed and revised the manuscript. Z.X. and Y.L. collected the samples and performed the tests. All authors approved the final manuscript as submitted and agree to be accountable for all aspects of the work.

## AUTHOR AFFILIATIONS

[1]Children's Hospital of Soochow University, Suzhou, Jiangsu, China
[2]Suzhou Industrial Park Center for Disease Control and Prevention, Suzhou, Jiangsu, China

## AUTHOR ORCIDs

Shenghao Hua http://orcid.org/0009-0000-4422-9139

## AUTHOR CONTRIBUTIONS

Daiyang Zhang, Conceptualization, Writing – original draft | Hewei Zhang, Data curation, Methodology | Xin Zhang, Funding acquisition, Project administration | Lei Yin, Funding acquisition, Project administration | Xuejun Shao, Methodology, Software,

Validation, Visualization | Shenghao Hua, Conceptualization, Data curation, Formal analysis, Methodology, Software, Writing – original draft, Writing – review and editing

## DATA AVAILABILITY

No data sets were generated during the current study.

## ETHICS APPROVAL

Our research received approval from the Ethics Board at the Children's Hospital of Soochow University, with approval number 2025CS133.

## ADDITIONAL FILES

The following material is available online.

### Open Peer Review

**PEER REVIEW HISTORY (review-history.pdf).** An accounting of the reviewer comments and feedback.

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
