## [Reviewer comments · Microbiology Spectrum]

Microbiology Spectrum

Identification of neutrophil-related genes associated with the severity of respiratory syncytial virus infection and their potential as biomarkers

Sheng-Hao Hua, Daiyang Zhang, Hwei Zhang, Xin Zhang, Lei Yin, and Xuejun Shao

Corresponding Author(s): Sheng-Hao Hua, Children's Hospital of Soochow University

Review Timeline:

Submission Date:	August 22, 2025
Editorial Decision:	November 3, 2025
Revision Received:	December 4, 2025
Editorial Decision:	January 5, 2026
Revision Received:	February 9, 2026
Accepted:	March 4, 2026

Editor: Elias Rahal

Reviewer(s): Disclosure of reviewer identity is with reference to reviewer comments included in decision letter(s). The following individuals involved in review of your submission have agreed to reveal their identity: Yoshitaka Kimura (Reviewer #2)

Transaction Report:

DOI: <https://doi.org/10.1128/spectrum.02591-25>

Re: Spectrum02591-25 (**Identification of neutrophil-related genes associated with the severity of respiratory syncytial virus infection and their potential as biomarkers**)

Dear Dr. Sheng-Hao Hua:

Thank you for the privilege of reviewing your work. Below you will find my comments, instructions from the Spectrum editorial office, and the reviewer comments.

Revision Guidelines

Sincerely,
Elias Rahal
Editor
Microbiology Spectrum

Reviewer #1 (Comments for the Author):

The manuscript presents a well-conducted study identifying key neutrophil-related genes linked to the severity of Respiratory Syncytial Virus (RSV) infection. The research employs robust bioinformatics analyses and verifies its findings with clinical samples, making a meaningful contribution to the understanding of RSV pathogenesis and potential biomarkers. The manuscript is well-written, clear, and effectively highlights the significance of the findings. However, a future direction towards protein

validation would significantly support the conclusions of this manuscript. Additionally, the figure titles and explanations should be refined to describe the figures accurately. Given its sound methodology and clinical relevance, it is suitable for publication and can move forward in the review process.

Reviewer #2 (Comments for the Author):

This study aimed to identify potential biomarkers associated with the severity of RSV infection, focusing on neutrophil-related genes, through analyses of public databases. The authors further validated the correlation between these candidate biomarkers and disease severity using clinical samples. This study is of interest to readers as it highlights novel potential biomarkers linked to neutrophil-related pathways. The limitations of this study are also discussed.

However, several aspects of this study warrant further consideration and clarification.

Major comments

#1. "Validation with clinical samples" in methods section (p 3, line 100-)

It is necessary to clearly describe the method of sample collection, including which medical institutions were involved, what type of patients were enrolled, the period during which the samples were collected, and the exclusion criteria applied in the sample collection process.

#2. "Validation of hub gene expression in GSE246622 and the predictive prognostic role of the hub genes" in results section (p 9, line 182-)

The results do not support the notion that candidate biomarkers have a predictive prognostic role. The authors showed that the expression levels of these genes decreased over time. However, I believe that this decrease, which is related to neutrophil activity or inflammation, may simply reflect the natural course of the disease rather than the prognosis of patients with RSV.

#3. "Diagnostic efficacy of the hub genes" in results section (p10, line 198-)

The procedures and evaluation methods of the ROC analysis were not clearly described. It should first be clarified which dataset was used for the evaluation.

The authors stated that they analyzed whether outpatients required hospitalization and whether hospitalized patients progressed to severe disease. However, it is necessary to organize and present the prognostic information of the original dataset used for the ROC analysis in the Results section. Specifically, please indicate how many outpatients were hospitalized and how many were not, as well as how many patients progressed to severe disease and how many did not.

In addition, are there any cases in the dataset used in which some prognostic details are missing? If such cases were excluded from the analysis, it should be clearly described in the Methods section.

#4. "Clinical validation" in results section (p11, line 215-)

Please specify the time points after disease onset at which the samples were collected for clinical validation and clarify whether there were any differences in the baseline characteristics among the study groups.

#5. "Clinical validation" in results section (p11, line 215-) and Fig. 5

I believe that these results may reflect neutrophil activation. However, did you evaluate the differential leukocyte count? In the severe group, was the proportion or absolute number of neutrophils significantly increased?

Minor comments

#1. Please clarify in the figure legends whether the bars represent mean {plus minus} SD, SEM, or percentiles in the box-and-whisker plots.

#2. In the clinical validation, the authors should use one-way ANOVA instead of t-test.

#3. Please describe what the samples are in GSE188427 and GSE236622 (e.g. whole blood cells or neutrophils).

Major comments

#1. "Validation with clinical samples" in methods section (p 3, line 100-)

It is necessary to clearly describe the method of sample collection, including which medical institutions were involved, what type of patients were enrolled, the period during which the samples were collected, and the exclusion criteria applied in the sample collection process.

We collected peripheral blood from children on the day they visited the Children's Hospital of Soochow University. These children exhibited typical respiratory symptoms and had positive nucleic acid tests for respiratory syncytial virus in their sputum or throat swabs. They were not diagnosed with hematological diseases, malignant tumors, or congenital heart disease.

#2. "Validation of hub gene expression in GSE246622 and the predictive prognostic role of the hub genes" in results section (p 9, line 182-)

The results do not support the notion that candidate biomarkers have a predictive prognostic role. The authors showed that the expression levels of these genes decreased over time. However, I believe that this decrease, which is related to neutrophil activity or inflammation, may simply reflect the natural course of the disease rather than the prognosis of patients with RSV.

After careful consideration, we also think that candidate biomarkers do not have a predictive prognostic role, so we revised the related statements in the article.

#3. "Diagnostic efficacy of the hub genes" in results section (p10, line 198-)

The procedures and evaluation methods of the ROC analysis were not clearly described. It should first be clarified which dataset was used for the evaluation.

The authors stated that they analyzed whether outpatients required hospitalization and whether hospitalized patients progressed to severe disease. However, it is necessary to organize and present the prognostic information of the original dataset used for the ROC analysis in the Results section. Specifically, please indicate how many outpatients were hospitalized and how many were not, as well as how many patients progressed to severe disease and how many did not.

In addition, are there any cases in the dataset used in which some prognostic details are missing? If such cases were excluded from the analysis, it should be clearly described in the Methods section.

The dataset GSE246622 was used for the evaluation. In GSE246622, we are unclear about how many outpatient patients were hospitalized, how many were not hospitalized, and how many patients progressed to severe disease. To address this, we plotted ROC curves using hub genes from 36 outpatient children and 58 hospitalized children, as well as from 18 critically ill children and 76 non-critically ill children in the same dataset. We determined the diagnostic efficacy of selected biomarkers by examining the expression levels of hub genes in children with known disease severity

#4. "Clinical validation" in results section (p11, line 215-)

Please specify the time points after disease onset at which the samples were collected for clinical validation and clarify whether there were any differences in the baseline characteristics among the study groups.

The time point for blood sampling is during the early stage of the disease, that is, 2-3 days after the child has symptoms of respiratory tract infection. The baseline characteristics among the groups are shown in the table below.

Characteristic	Disease severity			P
	Mild (n=31)	Moderate (n=27)	Severe (n=15)	
Sex, No. (%)				
Female	9 (29.0)	10 (37.0)	7 (46.7)	0.49
Male	22 (71.0)	17 (63.0)	8 (53.3)	
Age, median (IQR), mo	39.1 (24.0-48.0)	19.7 (6.5-36.0)	10.8 (2.5-11.0)	<0.05
Age, No. (%)				
<3 mo	0	1 (3.7)	4 (26.7)	
3 to <6 mo	1 (3.2)	4 (14.8)	2 (13.3)	
6 to <12 mo	1 (3.2)	6 (22.2)	5 (33.3)	
12 to <24 mo	3 (9.7)	3 (11.1)	1 (6.7)	
2 to <5 y	19 (61.3)	13 (48.1)	3 (20.0)	
5 to <18 y	7 (22.6)	0	0	

#5. "Clinical validation" in results section (p11, line 215-) and Fig. 5

I believe that these results may reflect neutrophil activation. However, did you evaluate the differential leukocyte count? In the severe group, was the proportion or absolute number of neutrophils significantly increased?

We conducted a retrospective analysis of the white blood cell classification counts of children in each group and found that the absolute number and percentage of neutrophils in the severe cases group did not significantly increase, as shown in the figure below.

Minor comments

#1. Please clarify in the figure legends whether the bars represent mean {plus minus} SD, SEM, or percentiles in the box-and-whisker plots.

The bars represent mean {plus minus} SD in Fig. 5. The upper line, middle line, and lower line in the box-and-whisker plots represent the upper quartile, mean, and lower quartile, respectively.

#2. In the clinical validation, the authors should use one-way ANOVA instead of t-test.

We have re-conducted the statistical analysis using one-way ANOVA instead of t-test.

#3. Please describe what the samples are in GSE188427 and GSE246622 (e.g. whole blood cells or neutrophils).

The samples are whole blood cells in GSE188427 and GSE246622.

Re: Spectrum02591-25R1 (**Identification of neutrophil-related genes associated with the severity of respiratory syncytial virus infection and their potential as biomarkers**)

Dear Dr. Sheng-Hao Hua:

Thank you for the privilege of reviewing your work. Below you will find my comments, instructions from the Spectrum editorial office, and the reviewer comments.

Revision Guidelines

Sincerely,
Elias Rahal
Editor
Microbiology Spectrum

Reviewer #1 (Comments for the Author):

The manuscript titled 'Identification of neutrophil-related genes associated with the severity of RSV infection' explores an important clinical area by identifying molecular biomarkers that correlate with disease progression in children. However, in its current form, the paper is primarily descriptive and lacks the depth of analysis. The methodology section requires more technical

detail, and the biological interpretation of the results remains surface-level. Furthermore, the redundancy between the identification of differentially expressed genes (DEGs) and pathway enrichment (KEGG) results suggests a need for better structural integration. Significant revisions are necessary to strengthen the scientific narrative and justify the clinical utility of the proposed biomarkers.

1. Clarification of ROC Curve Basis

The manuscript mentions using the `PROC` and `ggplot2` R packages for ROC analysis. However, it is not explicitly clear what specific clinical parameters or gene expression thresholds were used as the "gold standard" for the categories.

2. Insufficient Analytical Depth

The paper relies heavily on identifying hub genes and validating them via qRT-PCR. While this is a good start, modern bioinformatics papers require more advanced functional analysis.

Supplement your findings with additional analyses such as Gene Set Enrichment Analysis (GSEA) or Weighted Gene Co-expression Network Analysis (WGCNA). Instead of just showing that genes go up or down, analyze the networks and functional clusters they belong to.

3. Weak Methodology Section

The current methods are too brief. You mention using GEO2R for DEGs, but you do not describe the normalization steps or how you handled batch effects between the two different datasets (GSE188427 and GSE246622).

4. Merging DEGs and KEGG Sections

Currently, the results for DEGs and KEGG/GO enrichment are presented as separate, disconnected findings.

Merge these sections to create a more cohesive narrative. Instead of simply listing the DEGs and then listing the pathways, describe them together. For example: "Among the 117 neutrophil-associated DEGs identified, we observed a significant enrichment in pathways related to hematopoietic cell lineage and antigen processing (KEGG), further supporting the role of neutrophil activation in severe RSV".

Reviewer #2 (Comments for the Author):

The authors have satisfyingly answered all my questions.

Reviewer #1 (Comments for the Author):

The manuscript titled 'Identification of neutrophil-related genes associated with the severity of RSV infection' explores an important clinical area by identifying molecular biomarkers that correlate with disease progression in children. However, in its current form, the paper is primarily descriptive and lacks the depth of analysis. The methodology section requires more technical detail, and the biological interpretation of the results remains surface-level. Furthermore, the redundancy between the identification of differentially expressed genes (DEGs) and pathway enrichment (KEGG) results suggests a need for better structural integration. Significant revisions are necessary to strengthen the scientific narrative and justify the clinical utility of the proposed biomarkers.

1. Clarification of ROC Curve Basis

The manuscript mentions using the `PROC` and `ggplot2` R packages for ROC analysis. However, it is not explicitly clear what specific clinical parameters or gene expression thresholds were used as the "gold standard" for the categories.

ROC analysis uses the GSE246622 dataset, which categorizes RSV patients. These patients are classified into outpatients, mild hospitalized cases, and severe hospitalized cases, with clinical diagnosis serving as the gold standard for the different categories. However, the gene expression thresholds have not yet been determined in this study. This will be the focus of our future research.

2. Insufficient Analytical Depth

The paper relies heavily on identifying hub genes and validating them via qRT-PCR. While this is a good start, modern bioinformatics papers require more advanced functional analysis.

Supplement your findings with additional analyses such as Gene Set Enrichment Analysis (GSEA) or Weighted Gene Co-expression Network Analysis (WGCNA). Instead of just showing that genes go up or down, analyze the networks and functional clusters they belong to.

We sincerely appreciate the reviewer's insightful and constructive comments, which provide invaluable guidance for enhancing the functional and mechanistic interpretation of the neutrophil-related hub genes identified in our study. We fully concur that advanced bioinformatics analyses such as GSEA can better elucidate the functional clusters and regulatory networks associated with target genes, surpassing the basic characterization of individual gene expression changes. In accordance with the reviewer's suggestion, we have performed a comprehensive GSEA and incorporated all new findings into the revised manuscript.

3. Weak Methodology Section

The current methods are too brief. You mention using GEO2R for DEGs, but you do not describe the normalization steps or how you handled batch effects between the two different datasets (GSE188427 and GSE246622).

We appreciate the reviewer's valuable comment regarding the methodology. To clarify, the GSE188427 and GSE246622 datasets were analyzed independently with distinct purposes, thus eliminating the need for batch effect correction between them. For GSE188427, which was used to identify hub genes, raw data were processed via GEO2R, which employs quantile normalization as the default method to standardize gene expression values across samples within the dataset, ensuring consistency in gene expression measurements. Differential expression analysis (DEG) was then performed on this normalized dataset to compare RSV-infected inpatients, outpatients, and healthy controls, with significant DEGs defined by $|\log_2FC| > 0.5$ and adjusted $P < 0.05$. Neutrophil-related DEGs were further identified by intersecting these DEGs with neutrophil-related genes from Genecards, and hub genes were selected through PPI network analysis using the STRING database and Cytoscape. In contrast, GSE246622 was exclusively used for independent validation of the identified hub genes; prior to analysis, this dataset also underwent quantile normalization via GEO2R to normalize gene expression across its own samples (mild outpatients, moderate inpatients, and severe inpatients). The validation focused on comparing hub gene expression levels within GSE246622 to assess their correlation with disease severity, rather than merging or integrating data from the two datasets. Since the two datasets served separate discovery and validation roles without any cross-dataset integration, batch effects between them were not a relevant concern in this study.

4. Merging DEGs and KEGG Sections

Currently, the results for DEGs and KEGG/GO enrichment are presented as separate, disconnected findings.

Merge these sections to create a more cohesive narrative. Instead of simply listing the DEGs and then listing the pathways, describe them together. For example: "Among the 117 neutrophil-associated DEGs identified, we observed a significant enrichment in pathways related to hematopoietic cell lineage and antigen processing (KEGG), further supporting the role of neutrophil activation in severe RSV".

We appreciate the reviewer's insightful comment. We have merged the DEGs and GO/KEGG enrichment sections to form a more cohesive narrative, integrating the description of neutrophil-associated DEGs with their enriched biological functions and pathways to avoid disconnected findings. The merged description is shown below and has been reflected in the manuscript.

To identify neutrophil-related genes linked to RSV severity, we first analyzed the GSE188427 dataset encompassing healthy controls, RSV-infected outpatients, and RSV-infected inpatients. Using GEO2R with thresholds of adjusted $P < 0.05$ and $|\log_2FC| > 0.5$, we detected 392 upregulated and 98 downregulated genes in RSV-infected inpatients versus controls, and 459 upregulated and 134 downregulated genes in RSV-infected outpatients versus controls (Fig. 1A-B). Venn diagram analysis intersecting these DEGs with neutrophil-related genes retrieved from Genecards identified 117 neutrophil-associated DEGs exclusive to hospitalized pediatric RSV patients and not detected in outpatients (Fig. 1C). Heatmaps then visualized their distinct expression patterns across study groups (Fig. 1D), suggesting these genes may be uniquely implicated in severe disease phenotypes. Subsequently, we performed GO and KEGG enrichment analyses on these 117 neutrophil-associated DEGs to delineate their biological functions and associated pathways in RSV pathogenesis. Gene Ontology (GO) Biological Process (BP) enrichment revealed significant associations including the negative regulation of cytokine production, response to lipopolysaccharide, negative regulation of interleukin-6 production, and defense response to fungus (adjusted $P < 0.05$; Fig. 2A). These findings align with the well-established role of neutrophils in modulating inflammatory responses during viral respiratory infections and indicate that these neutrophil-associated DEGs may contribute to excessive or dysregulated inflammation in severe RSV. Cellular Component (CC) analysis showed enrichment in the cytoplasmic vesicle lumen, secretory granule lumen, and specific granules (Fig. 2A), which are subcellular compartments critical for neutrophil degranulation. This process is a key mechanism by which neutrophils release antimicrobial factors and proinflammatory mediators during infection. Molecular Function (MF) analysis indicated associations with immune receptor activity, MHC protein complex binding, and lipopolysaccharide binding (Fig. 2A), reflecting the involvement of these neutrophil-associated DEGs in neutrophil-mediated pathogen recognition and antigen presentation, processes central to antiviral immunity. Kyoto Encyclopedia of Genes and Genomes (KEGG) pathway enrichment analysis further revealed that the 117 neutrophil-associated DEGs were significantly enriched in pathways including hematopoietic cell lineage, antigen

processing and presentation, graft-versus-host disease, asthma, and drug metabolism-other enzymes (Fig. 2B). Enrichment in the hematopoietic cell lineage pathway supports the role of these genes in regulating neutrophil development and activation, processes known to be dysregulated in severe RSV infection. Meanwhile, enrichment in antigen processing and presentation underscores their potential involvement in modulating adaptive immune responses to RSV. Collectively, these functional and pathway analyses demonstrate that the 117 neutrophil-associated DEGs are biologically relevant to neutrophil-mediated inflammation and antiviral responses, highlighting their potential role as key mediators of RSV disease severity.

Re: Spectrum02591-25R2 (**Identification of neutrophil-related genes associated with the severity of respiratory syncytial virus infection and their potential as biomarkers**)

Dear Dr. Sheng-Hao Hua:

Your manuscript has been accepted, and I am forwarding it to the ASM production staff for publication. Your paper will first be checked to make sure all elements meet the technical requirements. ASM staff will contact you if anything needs to be revised before copyediting and production can begin. Otherwise, you will be notified when your proofs are ready to be viewed.

Sincerely,
Elias Rahal
Editor
Microbiology Spectrum

Reviewer #1 (Comments for the Author):

The revisions are satisfactory.

Reviewer #2 (Comments for the Author):

I am satisfied with the revisions that have been made by the authors.